Synthesis of light-inducible and light-independent anthocyanins regulated by specific genes in grape ‘Marselan’ (V. vinifera L.)

Ma Zong-Huan 1
Li Wen-Fang 1
Mao Juan 1
Li Wei 1
Zuo Cun-Wu 1
Zhao Xin 1
Dawuda Mohammed Mujitaba 1
Shi Xing-Yun 2
Chen Bai-Hong bhch@gsau.edu.cn 1
1 College of Horticulture, Gansu Agricultural University , Lanzhou , China
2 Wuwei Academy of Forestry Science , Wuwei , China
Tiessen Axel
Electronic publication date: 2019 Mar 1
Publication date: 2019
Volume: 7
Electronic Location ID: e6521
Received 2018 Dec 3; Accepted 2019 Jan 24
Copyright: ©2019 Ma et al.
Copyright year: 2019
Copyright holder: Ma et al.
License: This is an open access article distributed under the terms of the Creative Commons Attribution License, which permits unrestricted use, distribution, reproduction and adaptation in any medium and for any purpose provided that it is properly attributed. For attribution, the original author(s), title, publication source (PeerJ) and either DOI or URL of the article must be cited.
License URL: https://creativecommons.org/licenses/by/4.0/

Keywords: Grapevine, Transcriptome analysis, Anthocyanin composition, Light intensity, LC-MS/MS

Funding: National Natural Science Foundation of China 31460499 Science and Technology Major Project of Gansu Province 18ZD2NA006 Forestry Science and Technology Plan of Gansu Province 2015kj023 This work was supported by the National Natural Science Foundation of China (31460499), the Science and Technology Major Project of Gansu Province (18ZD2NA006) and the Forestry Science and Technology Plan of Gansu Province (2015kj023). The funders had no role in study design, data collection and analysis, decision to publish, or preparation of the manuscript.

==============================
Anthocyanin is an important parameter for evaluating the quality of wine grapes. However, the effects of different light intensities on anthocyanin synthesis in grape berry skin and its regulation mechanisms are still unclear. In this experiment, clusters of wine grape cv. ‘Marselan’ were bagged using fruit bags with different light transmittance of 50%, 15%, 5%, and 0, designated as treatment A, B, C and D, respectively. Fruits that were not bagged were used as the control, designated as CK. The anthocyanin composition and concentration, as well as gene expression profiles in the berry skin were determined. The results showed that the degree of coloration of the berry skin reduced with the decrease of the light transmittance, and the veraison was postponed for 10 days in D when compared with the CK. Total anthocyanin concentration in the berry skin treated with D decreased by 51.50% compared with CK at the harvest stage. A total of 24 and 21 anthocyanins were detected in CK and D, respectively. Among them, Malvidin-3-O-coumaroylglucoside (trans), which showed a significant positive correlation with the total concentration of anthocyanins at the harvest stage (r = 0.775) and was not detected in D, was presumed to be light-induced anthocyanin. Other anthocyanins which were both synthesized in CK and D were considered to be light-independent anthocyanins. Among them, Malvidin-3-O-coumaroylglucoside (cis) and Malvidin-3-O-acetylglucoside were typical representatives. Remarkably, the synthesis of light-inducible anthocyanins and light-independent anthocyanins were regulated by different candidate structural genes involved in flavonoid biosynthesis pathway and members of MYB and bHLH transcription factors.

Introduction

Phenolic compounds play an important role in the sensory properties of wine (Quijada-Morín et al., 2012), and can be divided into two categories, flavonoids and nonflavonoids (Figueiredo-González et al., 2012a; Figueiredo-González et al., 2012b; Figueiredo-González et al., 2012c; Pérez-Gregorio et al., 2014; Perez-Gregorio & Simal-Gandara, 2017a; Perez-Gregorio & Simal-Gandara, 2017b). Anthocyanin is one of the important flavonoids and plays an important role in the formation of fruit color and quality (Bellincontro et al., 2010; Figueiredo-González et al., 2012a; Figueiredo-González et al., 2012b; Figueiredo-González et al., 2012c; Ilieva et al., 2016; Moro, Hassimotto & Purgatto, 2017). The concentration and composition of anthocyanins during fruit ripening are related to the climate, soil, place of growing and cultivation (Ortega-Regules et al., 2006; Carbone et al., 2009; Figueiredo-González et al., 2013; Liang et al., 2014; Zhang et al., 2018a; Zhang et al., 2018b). Light is an important environmental factor affecting the synthesis of flavonoids in most plants (Guan et al., in press; He et al., 2016). Therefore, it is important to study the effects of light and other environmental factors on anthocyanin synthesis and reveal the regulation mechanism for improving fruit quality.

Studies have shown that higher light intensity can promote the accumulation of anthocyanins in most plants (Maier & Hoecker, 2015). In ‘Hakuho’ peach (Prunus persica Batsch), the area and intensity of the skin’s red color were increased with increasing bag exposure to sunlight (Jia, Araki & Okamoto, 2005). In addition to light intensity, light quality can also affect the synthesis of anthocyanins. The previous studies showed that the anthocyanin concentrations under red and blue light treatment were higher than that in the dark conditions (Xu et al., 2014; Liu et al., 2015). UV-light alters concentrations of individual anthocyanins in grape berry skin (Crupi et al., 2013; Zoratti et al., 2014). In red pear cv. ‘Red Zaosu’, continuous illumination with blue light and red light increased the concentration of anthocyanins in the pericarp, and the blue light promoted the accumulation of anthocyanins more than the red light (Tao et al., 2018). The concentration and proportion of anthocyanin in strawberries (Fragaria ×ananassa) treated with colored light-quality selective plastic films were significantly different from those of the control (Miao et al., 2016). Among these, the red and yellow films promoted anthocyanin accumulation, while the green and blue films decreased anthocyanin concentration.

The anthocyanin composition in the grape berry skin is also affected by cultivar, temperature and light conditions by altering the expression of genes involved in the flavonoid biosynthetic pathway (Azuma et al., 2012). Concentrations of total anthocyanins, pelargonidin-3-glucoside and pelargonidin-3-malonylglucoside in strawberries were significantly increased after blue and red light treatment (Zhang et al., 2018a). In the veraison stage of grapes, the formation of color is mainly due to the synthesis of malvidin-based, and the anthocyanin concentration in grape berries at mature stage was significantly higher than that of the control after blue and red light irradiation (Kondo et al., 2014). Some studies suggested that the expression of flavanone 3′-hydroxylase (F3′H), flavonoid 3′5′ hydroxylase (F3′5′H) and O-methyltransferases (OMT) strongly affected the variation in anthocyanin composition (Castellarin & Gaspero, 2007; Azuma et al., 2012). In the berry skin of grape cv. ‘Guipu No. 6’, the proportion of cyanidin-type (3′-substituted) anthocyanins was higher than that in ‘Cabernet Sauvignon’, which could be related to higher expression of F3′H (Cheng et al., 2017). F3′5′H is the key enzyme in delphinidin biosynthesis (Fukui et al., 2003), whereas F3′H catalyzes the metabolism of naringenin to eriodictyol (Doostdar et al., 1995; Zhou et al., 2016). Light exclusion increased the ratio of dihydroxylated to trihydroxylated anthocyanins in grape, which in parallel with F3′H and F3′5′H transcript amounts (Guan et al., in press). Transcription factors encoding MYB-like and bHLH proteins appeared to modulate the expression of the structural genes involved in anthocyanin biosynthesis pathway (Lepiniec et al., 2006; Azuma et al., 2012). The apple MdMYB1/MdMYBA transcription factor has been shown in earlier studies to be a positive regulator of light-controlled anthocyanin biosynthesis (Takos et al., 2006a; Takos et al., 2006b; Ban et al., 2007; Lin-Wang et al., 2010). A new study found that MdMYB1 accumulates in light, but is degraded via a ubiquitindependent pathway in the dark (Li et al., 2012). Cominelli et al. (2008) reported that bHLH genes analysed showed light induction, and their expression preceded that of the late structural genes, suggesting their possible role in light regulation of these structural genes (Cominelli et al., 2008).

Bagged apple fruits cannot be colored, and the removal of the fruit bags resulted in a rapid increase in anthocyanin concentration in the skin, which became red in a relatively short period of time (Wang, Wei & Ma, 2015). The leaf color of red lettuce (Lactuca sativa L.) developed poorly coloring grown at low light intensity (Zhang et al., 2018b). However, several studies suggested that anthocyanin biosynthesis was not readily affected by sunlight in grape (Price et al., 1995; Downey, Harvey & Robinson, 2010). In addition, the difference of anthocyanin components in grape under different light intensity is still not clear. This study is based on the hypothesis that light-independent anthocyanins are dominant in grapes, and their synthesis is regulated by different genes. In this experiment, we analyzed the composition of anthocyanin in the berry skin and the key genes in its metabolic pathway under fruit bags with different light transmission properties to clarify the role of light in the regulation of anthocyanin synthesis in wine grape cv. ‘Marselan’.

Materials & Methods

Plant materials and experimental design

A nine-year-old wine grape cv. ‘Marselan’ orchard belonging to the Wuwei Academy of Forestry Science (102°42′E, 38°02′N; altitude 1,632 m; Gansu, China) was used for the present study. The climate in this area corresponds to the warm temperate zone, with an average annual temperature of 6.9 °C, an average annual rainfall of 191 mm, an annual evaporation of 2,130 mm and an sunshine duration of 2,720 h. The frost-free period is more than 160 days, with long duration of sunshine and large variations in temperature during production. The grape plants were self-rooted and planted at 3 × 0.5 m spacing. Seven adjacent vine rows, each containing approximately 120 vines, were used for the experiment. Five were manipulated and measured, and two were used as border rows, positioned between of experimental rows. Each plant was pruned to allow only five branches and we also prune to allow two clusters per branch. Drip irrigation with the volume of 4,800 m3 ha−1 was applied. In addition, N, P2O5 and K2O was applied to 160 kg ha−1, 120 kg ha−1 and 240 kg ha−1, respectively. Clusters were bagged at 45 days after flowering and the bags were maintained until harvest. Four different bags (wood pulp, Zhaofeng, Yantai, China), including one-layered white bags, stripe (yellow and brown) bags, brown bags and two-layered bags (the inner layer was completely red and the outer bag had black inner lining whiles the outside of the outer bag was brown), designated as A, B, C and D, with light transmittance of 50%, 15%, 5% and 0, respectively. The control (CK) treatment consisted of fruits that were not bagged. Each of the five treatments had 90 clusters from 30 grapevines, and each treatment was replicated three times. Berries were harvested at 90 (S1), 100 (S2) and 125 (S3) days after flowering, respectively. Each repetition was harvested from three clusters, of which 10 berries were randomly selected from the base, middle and top of the cluster, respectively. A total of 30 berries were randomly selected from each replicate. Each treatment was replicated three times. The berry skin was quickly peeled off, immediately frozen in liquid nitrogen and stored at −80 °C for further analysis of anthocyanin contents and composition. In addition, samples from S2 (CK1, A1, B1, C1 and D1) and S3 (CK2, A2, B2, C2 and D2) were used for RNA-seq analysis.

Ultra-high performance liquid chromatography-tandem mass spectrometry (UHPLC–MS/MS) analysis of anthocyanins

Total anthocyanin concentration was analyzed from each treatment according to the technique described (Rabino & Mancinelli, 1986) with some modifications. In brief, lyophilized berry skin samples were finely ground with liquid nitrogen, and 1.0 g of powdered samples were transferred to the extract solution (20 mL) containing methanol/hydrochloric acid mixture (99:1 v/v) to extract anthocyanins at 4 °C for 24 hours. The mixture was incubated and then centrifuged at 9,168 g at 4 °C for 20 min. The supernatant was transferred into the cuvette, and it concentrations at 530 nm and 657 nm were determined by spectrophotometer (Shimadzu, Kyoto, Japan). A = OD530- 0.25 OD657 was used to calculate the concentration of anthocyanins.

The extraction of the total anthocyanin component was done as described by He et al. (2016) with some modifications. The berry skin is placed in a mortar and ground with liquid nitrogen. About 2.0 g of the powdered samples were added in a 10 mL centrifuge tube with 8 mL 2% formic acid-methanol solution. After 10 min of ultrasonic oscillation, the extract was placed in the dark at 25 °C on 4 g for 30 min, and centrifuged at 4 °C on 13,201 g for 10 min. The supernatant fraction was changed into a 50 mL centrifuge tube and the residue was re-extracted three times. The vacuum rotary evaporator (BUCHI, New Castle, DE, USA) was used to vaporize the organic fraction at 40 °C. The residual parts were poured into the active solid phase extraction cartridge (Phenomenex, Macclesfield, UK). The residue was flushed twice with 5 mL water. After removing the leachate, the solid phase extraction cartridge was eluted twice with 10 mL methanol. The filtrate was collected and evaporated. Malvidin-3-O-glucoside as external standard quantitative, mass concentration gradient was 0–50 mg L−1; other anthocyanins were calculated to be equivalent to the concentration of Malvidin-3-O-glucoside.

UHPLC analysis was performed on an Agilent 1290 UHPLC system coupled with an Agilent 6460 triple quadrupole mass spectrometer in positive ion mode ([M+H]+). The chromatographic column was 120 EC-C18 column (150 × 2.1 mm, 2. 7μm, 40 °C.). The mobile phases were filtered with 0. 45μm degassed membrane filter in vacuum. Phase A (formic acid : water was 0.5: 100 v/v) and B (formic acid : methanol: acetonitrile was 0.5 : 50 : 50, v/v). The gradient elution: 0 min, 90% of A, 10% of B; 0–28 min, 54% of A, 46% of B; 28–29 min, 90% of A, 10% of B; 29–34 min, 90% of A, 10% of B. The sample volume was 3 µL, the flow rate was 0.4 mL/minute, and a column temperature was 30 °C. Mass spectrometry results were analyzed using Mass Lynx V4.1 software, and the identification of the material structure was referenced to the standard substance as described by Lopes-da Silva et al. (2002).

RNA isolation, cDNA library construction and transcriptome sequencing

The total RNA was extracted with RNA plant reagent (Real-Times Biotechnology, Beijing, China) and evaluated with a 1% agarose gel stained with GoldView. The RNA quality and quantity were assessed by using a Nanodrop 2000 Spectrophotometer and an Agilent 2100 Bioanalyzer (Agilent Technologies, Sangta Clara, CA, USA). For each berry skin treatment, the RNA specimen of three randomly sampled individuals were aggregated as mixed samples. These mixed samples were used for cDNA construction and RNA sequencing, which were completed by Guangzhou Sagene Bioinformation Technology Co. Ltd. The cDNA library was constructed with 3 µg RNA of each sample. To select DNA fragments of preferentially 150–200 bp in length, library fragments were purified with AMPure XP system (Beckman Coulter, Beverly, MA, USA). Other details for the process of library construction was described in Mao et al. (2018), and the 10 libraries were sequenced on an Illumina HiSeq 2000 platform.

RNA-seq data analysis

Raw reads were cleaned by removing the adapter sequences and the clean reads were aligned to the grape reference genome (http://plants.ensembl.org/Vitis_vinifera/Info/Index) using the program Tophat v2.0.943. During the detection process of differentially expressed genes (DEGs), the absolute value of the log2 (fold change) with Fragments Per Kilobase Million (FPKM) ≥ 1 were used as the threshold to determine the significantly DEGs in this research. The DEGs were analyzed by GOseq R software packages (Mao et al., 2018) and had a significant enrichment effect on the modified Gene Ontology (GO) terms with corrected P value <0.05. In order to get the detailed function of classification in different treatments, software KOBAS was used to conduct a full enrichment test of different genes expressed in the KEGG pathway. The pathways with Q value ≤ 0.01 were considered significantly enriched (Kanehisa et al., 2004; Xie et al., 2011).

Quantitative real-time PCR validation of RNA-seq data

To quantitatively determine the reliability of our transcriptional data, we monitored the expression of four candidate DEGs using qRT-PCR. Specific primer pairs were designed as shown in Table S1. The qRT-PCR was performed using the Roche Light Cycler96 Real-Time Detection System (Roche, Basel, Switzerland) with SYBR Green PCR Master Mix (Takara, Kusatsu, Japan). The thermal profile for SYBR Green I RT-PCR was 95 °C for 15 min, followed by 40 cycles of 95 °C for 10s and 60 °C for 30s and 72 °C for 30s. The reference gene UBI (XM_002266714) was used as internal reference. The comparative 2−ΔΔCT method was used to analyze the expression levels of the different genes (Livak & Schmittgen, 2001). All of the samples were tested in triplicate, and the experiments were performed on three biological replicates.

Statistical analysis

Statistical analyses were performed using analysis of variance (ANOVA) followed by Duncan’s new multiple range tests with SPSS version 17.0 (SPSS, Chicago, IL, USA). A significance level of p <0.05 was applied. Correlation tests were performed using Pearson product-moment correlation coeffcient (Pearson’s r) with a two-tailed test.

Results

Effects of light intensity on veraison and total anthocyanin concentration in grape berry skin

Results showed that different bagging treatments affected duration of color change in grape berries (Fig. 1A). The veraison of A and CK occurred almost simultaneously, but the coloring of B was 3 days later compared with CK. while that of C and D was delayed for 10 days. The color conversion rates of CK, A and B were higher than 50% at S1, but there was no coloring in the clusters of C and D. In addition, the clusters of CK and A were completely coloured at S2, while C and D were just beginning to be coloured. However, berries from all treatments were completely coloured and the appearance was not different at S3. The total anthocyanin concentration decreased with decreases in light transmittance of fruit bags from the three development stages, of which A, B, C and D were decreased by 10.10%, 19.23%, 37.07% and 51.50% compared with that of CK at the harvest stage, respectively (Fig. 1B). Summing up the above, the samples of S2 and S3 were selected for RNA-seq analysis.

Effects of light intensity on anthocyanin composition in grape berry skin

UHPLC-MS/MS was used to detect anthocyanin composition in the berry skin at three different developmental stages, and a total of 24 anthocyanin components were obtained (Figs. 2A– 2C). Among them, these 24 anthocyanins were present in both CK and A at the S3, and only Cyanidin-3-O-coumaroylglucoside (trans) was not detected in B at S2 and S3 but synthesized at S1. In addition, Cyanidin-3-O-coumaroylglucoside (cis) and Dephinidin-3-O-coumaroylglucoside (cis) were not detected in C at S3 while Dephinidin-3-O-coumaroylglucoside (cis) was detected at S1. Furthermore, Cyanidin-3-O-coumaroylglucoside (cis), Peonidin-3-O-coumaroylglucoside (trans) and Malvidin-3-O-coumaroylglucoside (trans) were not detected in D at S1, S2 and S3, whereas the concentration of Cyanidin-3-O-coumaroylglucoside (cis) was very low in CK and other bagging treatments from three development stages (0.5–11.5 mg kg−1). Therefore, these results initially indicated that Peonidin-3-O-coumaroylglucoside (trans) and Malvidin-3-O-coumaroylglucoside (trans) may be induced by light intensity, and also suggested that the composition of anthocyanins decreased as the light intensity was decreased.

Figure 1 Effects of different fruit bags on berry skin color.

Bags with light transmittance of 50%, 15%, 5% and 0 were performed at 45 days after flowering and designated as CK, A, B, C and D, respectively. Unbagged berries as a control . (A) Close-up views of wine grape cv. ‘Marselan’ berry fruit from different bagging treatments at S1, S2 and S3. (B) Changes of a nthocyanin concentration at the three development stages. Error bars represent the ±SE of three biological replicates, and the asterisks represent significant differences from the control, with P < 0.05 (*) or P < 0.01(∗∗).

Figure 2 Heatmap of the concentration of anthocyanins in different bagging treatments at S1, S2 and S3.

Correlation and proportion analysis of individual anthocyanins and total anthocyanins in the berry skin

Anthocyanin concentrations from the harvest stage were chosen for subsequent analysis since all the bagged fruits were completely colored during this period. The concentration of Peonidin-3-O-coumaroylglucoside (trans), Petunidin-3-O-coumaroylglucoside (cis) and Malvidin-3-O-coumaroylglucoside (trans) positively correlated (r = 0.775–0.892) with the total anthocyanin concentration after bagging treatments (Data S1). However, the proportions of Peonidin-3-O-coumaroylglucoside (trans) and Petunidin-3-O-coumaroylglucoside (cis) in CK were 1% and 0, respectively (Fig. 3A). Similarly, they were 0 in D (Fig. 3B). The proportion of Malvidin-3-O-coumaroylglucoside (trans) was 12% in CK and was 0 in D. These results further indicated that Malvidin-3-O-coumaroylglucoside (trans) was induced by light intensity. Nevertheless, the proportion of Malvidin-3-O-coumaroylglucoside (cis) was 14% in D and 1% in CK. The concentration of Malvidin-3-O-acetylglucoside was highest in both CK and D, which were 27% and 34%, respectively. Therefore, except for light-inducible anthocyanin Malvidin-3-O-coumaroylglucoside (trans), other anthocyanins which were both synthesized in CK and D were considered to be light-independent anthocyanins, especially Malvidin-3-O-coumaroylglucoside (cis) and Malvidin-3-O-acetylglucoside.

Figure 3 The proportion of individual anthocyanins in total anthocyanin concentration from CK (A) and D (B) treatments at S3.

Comparative transcriptome analysis identified key processes and genes responsible for anthocyanin accumulation regulated by light intensity

In order to further elucidate the molecular basis for anthocyanin accumulation, gene expression profiles in different light intensities were analyzed by comparative transcriptomic sequencing. After the sequencing quality control, 5.96 Gb clean bases were generated from the ten libraries, and the Q30 base percentage of each sample was not less than 90.35% (Table S2). Moreover, 97.77%–99.19% of high-quality 150 bp reads were selected for further analysis (Table S3). Clean reads of the various samples were aligned against the specified grape reference genome. Mapped reads ranged from 46.36% to 78.43%. Uniquely mapped reads and multiple- mapped reads were ranged from 46.10% to 78.00% and 0.26% to 0.43%, respectively.

Then, changes in gene expression levels were determined by comparing each of the treatment with the CK at S2 and S3, which was CK1 versus A1, CK1 versus B1, CK1 versus C1, CK1 versus D1 and CK2 versus A2, CK2 versus B2, CK2 versus C2, CK2 versus D2, respectively. At S2, the down-regulated genes increased from 502 to 1,482, and the up-regulated genes increased from 721 to 2,308 with the decrease of light intensity compared with CK. At S3, the down-regulated genes increased from 1,592 to 2,429, whiles the up-regulated genes increased from 1,326 to 1,599 with decreases in light intensity compared with CK (Fig. S1). GO and KEGG enrichment analyses were conducted to understand the functions of DEGs using all reference genes as background. GO term enrichment analysis categorized the annotated sequences into three main categories: biological process, cellular component and molecular function (Table S4). In the biological process category, the terms “response to biotic stimulus” and “response to endogenous stimulus” was shared in CK1 versus B1, CK1 versus C1, CK1 versus D1, CK2 versus A2, CK2 versus B2, CK2 versus C2 and CK2 versus D2, except for in CK1 versus A1. The terms “response to stress” were shared in CK1 versus A1, CK1 versus B1, CK1 versus C1, CK1 versus D1, CK2 versus A2, CK2 versus B2, CK2 versus C2 and CK2 versus D2. The terms “signal transduction”, “regulation of cellular process” and “cellular response to stimulus” were shared in CK1 versus B1, CK1 versus C1, CK1 versus D1, CK2 versus A2, CK2 versus B2 and CK2 versus D2. In the cellular component category, the terms “external encapsulating structure”, “cell wall”, “membrane” and “cell periphery” were shared in CK1 versus A1, CK1 versus B1, CK1 versus C1, CK1 versus D1, CK2 versus A2, CK2 versus B2, CK2 versus C2 and CK2 versus D2. In the molecular function category, the DEGs were further classified into 13 major groups in CK2 versus B2 and CK2 versus C2, 11 in CK2 versus D2, 10 in CK1 versus D1, nine in CK1 versus B1 and CK2 versus A2, six in CK1 versus A1, four in CK1 versus C1. Furthermore, KEGG pathways, including flavonoid biosynthesis, circadian rhythm-plant, stilbenoid, diarylheptanoid and gingerol biosynthesis and phenylalanine metabolism were significantly enriched in CK1 versus A1, CK1 versus B1, CK1 versus C1, CK1 versus D1, CK2 versus A2, CK2 versus B2, CK2 versus C2 and CK2 versus D2 (Table 1). Among these, flavonoid biosynthesis pathway was selected for subsequent analysis.

Table 1 Significantly enriched pathways of DEGs between different treatments.

Pathway ID	Pathway	Number of DEGs	Background number	Q-value (<0.01)	
	CK1-vs-A1				
ko00941	Flavonoid biosynthesis	39	73	0.000000	
ko04712	Circadian rhythm - plant	28	60	0.000000	
ko00945	Stilbenoid, diarylheptanoid and gingerol biosynthesis	13	23	0.000000	
ko00360	Phenylalanine metabolism	19	57	0.000000	
ko00940	Phenylpropanoid biosynthesis	22	172	0.002593	
	CK1-vs-B1				
ko00196	Photosynthesis - antenna proteins	11	19	0.000008	
ko04626	Plant-pathogen interaction	34	171	0.000138	
ko00941	Flavonoid biosynthesis	17	73	0.004055	
ko00073	Cutin, suberine and wax biosynthesis	9	25	0.004055	
ko04075	Plant hormone signal transduction	34	214	0.007062	
	CK1 versus C1				
ko00941	Flavonoid biosynthesis	18	73	0.000021	
ko04626	Plant-pathogen interaction	27	171	0.000170	
	CK1 versus D1				
ko00941	Flavonoid biosynthesis	52	73	0.000000	
ko04712	Circadian rhythm - plant	33	60	0.000000	
ko00944	Flavone and flavonol biosynthesis	15	21	0.000001	
ko00360	Phenylalanine metabolism	26	57	0.000004	
ko00350	Tyrosine metabolism	21	44	0.000019	
ko00945	Stilbenoid, diarylheptanoid and gingerol biosynthesis	14	23	0.000027	
ko00710	Carbon fixation in photosynthetic organisms	25	62	0.000062	
ko01230	Biosynthesis of amino acids	58	206	0.000070	
ko00010	Glycolysis / Gluconeogenesis	36	110	0.000124	
ko01200	Carbon metabolism	63	238	0.000197	
ko00250	Alanine, aspartate and glutamate metabolism	17	43	0.001956	
ko00196	Photosynthesis - antenna proteins	10	19	0.002533	
ko00260	Glycine, serine and threonine metabolism	22	66	0.003610	
ko00750	Vitamin B6 metabolism	8	14	0.004533	
ko00950	Isoquinoline alkaloid biosynthesis	11	24	0.004618	
ko04626	Plant-pathogen interaction	44	171	0.004796	
ko00052	Galactose metabolism	19	58	0.008664	
ko00960	Tropane, piperidine and pyridine alkaloid biosynthesis	12	30	0.009569	
	CK2 vs-A2				
ko00941	Flavonoid biosynthesis	43	73	0.000000	
ko04712	Circadian rhythm - plant	30	60	0.000000	
ko00195	Photosynthesis	26	52	0.000000	
ko00196	Photosynthesis - antenna proteins	14	19	0.000000	
ko00945	Stilbenoid, diarylheptanoid and gingerol biosynthesis	15	23	0.000000	
ko00940	Phenylpropanoid biosynthesis	50	172	0.000000	
ko00360	Phenylalanine metabolism	23	57	0.000003	
ko00592	alpha-Linolenic acid metabolism	17	56	0.007642	
	CK2 vs-B2				
ko00941	Flavonoid biosynthesis	40	73	0.000000	
ko04712	Circadian rhythm - plant	29	60	0.000000	
ko00360	Phenylalanine metabolism	24	57	0.000000	
ko00945	Stilbenoid, diarylheptanoid and gingerol biosynthesis	14	23	0.000000	
ko00940	Phenylpropanoid biosynthesis	40	172	0.000020	
ko04626	Plant-pathogen interaction	36	171	0.000659	
ko00130	Ubiquinone and other terpenoid-quinone biosynthesis	12	36	0.003511	
ko00592	alpha-Linolenic acid metabolism	15	56	0.007769	
	CK2 vs-C2				
ko00941	Flavonoid biosynthesis	37	73	0.000000	
ko04712	Circadian rhythm - plant	28	60	0.000000	
ko00945	Stilbenoid, diarylheptanoid and gingerol biosynthesis	14	23	0.000000	
ko00360	Phenylalanine metabolism	21	57	0.000001	
ko00940	Phenylpropanoid biosynthesis	38	172	0.000013	
ko04626	Plant-pathogen interaction	35	171	0.000194	
	CK2-vs-D2				
ko00941	Flavonoid biosynthesis	45	73	0.000000	
ko00360	Phenylalanine metabolism	32	57	0.000000	
ko00195	Photosynthesis	30	52	0.000000	
ko04712	Circadian rhythm - plant	32	60	0.000000	
ko00945	Stilbenoid, diarylheptanoid and gingerol biosynthesis	15	23	0.000015	
ko00940	Phenylpropanoid biosynthesis	54	172	0.000136	
ko00196	Photosynthesis - antenna proteins	12	19	0.000240	
ko00592	alpha-Linolenic acid metabolism	21	56	0.005319	
ko00400	Phenylalanine, tyrosine and tryptophan biosynthesis	17	43	0.008449	

Genes involved in flavonoid biosynthesis pathway

The expression levels of anthocyanin biosynthesis related genes were significantly different among the different treatments. The expression levels of phenylalanine ammonium lyases (PALs; VIT_06s0004g02620, VIT_13s0019g04460, VIT_16s0039g01360, VIT_00s2849g00010, VIT_16s0039g01280 and VIT_08s0040g01710), trans-cinnamate 4-monooxygenase-like (C4H; VIT_06s0004g08150), CHS (VIT_16s0022g01190, VIT_16s0022g01140, VIT_14s0068g00920, VIT_14s0068g00930 and VIT_05s0136g00260), chalcone isomerase (CHI; VIT_13s0067g03820), F3′5′Hs/F3′Hs (VIT_06s0009g02970, VIT_06s0009g02840 and VIT_06s0009g02860, VIT_17s0000g07210), flavanone 3-hydroxylase (F3H; VIT_04s0023g03370), DFR (VIT_18s0001g12800) and LDOX (VIT_02s0025g04720) in D were significantly lower than that of CK at S2. The expression levels of F3′5′Hs/F3′H s (VIT_06s0009g02970, VIT_06s0009g02840 and VIT_17s0000g07210), F3H (VIT_04s0023g03370), and LDOX (VIT_02s0025g04720) were significantly up-regulated compared with CK at S3. At S3, the expression levels of PALs (VIT_13s0019g04460, VIT_16s0039g01360, VIT_00s2849g00010, VIT_16s0039g01280, VIT_16s0039g01240, VIT_16s0039g01300 and VIT_08s0040g01710) and C4H (VIT_06s0004g08150) in A2, B2 and D2 were down-regulated compared with that of CK2 (Fig. 4).

Figure 4 Heatmap of expressed genes assigned to anthocyanins synthesis in different bagging treatments.

Colors indicate expression values of the genes. Expression values of ten libraries are presented as FPKM normalized log2 transformed counts.

Regulation of MYB transcription factors by different light intensities

At S3, MYB1R1 (VIT_00s0299g00060), MYB4B (VIT_04s0023g03710), MYB15 (VIT_05s0049g01020), MYBB1R1 (VIT_17s0000g07510), MYB44 (VIT_18s0001g09850), MYB80 (VIT_19s0015g01280) and MYB-like protein H isoform X1 (XLOC_013017) were up-regulated in bagged berry skin, of which MYB4B (VIT_04s0023g03710) and MYB15 (VIT_05s0049g01020) were most up-regulated in D treatment. However, MYB5B (VIT_06s0004g00570), MYB108-like protein 2 (VIT_07s0005g01950), MYBCS1 (VIT_08s0007g07230), MYB60 (VIT_08s0056g00800) and PHL6 (VIT_09s0054g01620) were significantly down-regulated (Fig. 5A).

Figure 5 Heatmap of MYB and basic helix-loop-helix (bHLH) transcription factors in different bagging treatments.

Colors indicate expression values of the genes. Expression values of ten libraries are presented as FPKM normalized log2 transformed counts.

The expression of bHLH47 (VIT_14s0108g00480), MYC2 (VIT_15s0046g00320), bHLH137-like (VIT_17s0000g00430), bHLH79 (VIT_17s0000g05370), bHLH93 (VIT_18s0001g08040) and bHLH123 (VIT_19s0014g04670) in bagging treatments were significantly inhibited compared with that of CK at S3. bHLH147 (VIT_05s0077g00750), MYC1 (VIT_07s0104g00090), bHLH106 (VIT_08s0040g01240), bHLH68 (VIT_09s0002g04120, VIT_11s0016g03560), bHLH144 (VIT_10s0003g02940), bHLH41 (VIT_11s0016g02070), bHLH30 (VIT_12s0028g02350), bHLH36 (VIT_12s0028g03550) and bHLH48 (VIT_08s0007g07870) were up-regulated in bagging treatments (Fig. 5B).

Light-inducible and light-independent anthocyanins regulated by key candidate genes

The concentration of light-inducible anthocyanin Malvidin-3-O-coumaroylglucoside (trans) was selected for the correlation analysis with genes involved in flavonoid biosynthesis pathway. In addition, although Malvidin-3-O-acetylglucoside accounted for the largest proportion of anthocyanin synthesis, Malvidin-3-O-coumaroylglucoside (cis) had the largest difference between CK and D. The reason was that Malvidin-3-O-coumaroylglucoside (cis) was an isomer of Malvidin-3-O-coumaroylglucoside (trans) and was therefore selected for correlation analysis. The concentration of Malvidin-3-O-coumaroylglucoside (trans) positively correlated with the expression of PALs (VIT_16s0039g01360, VIT_00s2849g00010, VIT_16s0039g01240, VIT_08s0040g01710, VIT_16s0039g01300; r = 0.773–0.795) and F3H (VIT_18s0001g14310; r = 0.887), but it negatively correlated with CHS (VIT_14s0068g00920), F3H (VIT_04s0023g03370) and F3′5′H (VIT_06s0009g02970) (r =  − 0.973—0.796) (Data S2). The concentration of Malvidin-3-O-coumaroylglucoside (cis) positively correlated with the expression level of F3′5′H (VIT_06s0009g02840; r = 0.923) but it negatively correlated with CHS (VIT_05s0136g00260; r =  − 0.749).

For transcription factors, light-inducible anthocyanin synthesis positively correlated with MYB members (VIT_01s0011g04760, MYBC2-L1; VIT_02s0033g00390, MYBA2; VIT_02s0033g00410, MYBA1; VIT_02s0033g00450, MYBA3; VIT_17s0000g06190, MYB30; r = 0.702–0.812) and bHLH (VIT_03s0038g01790, bHLH121; VIT_12s0028g02350, bHLH30; VIT_17s0000g05370, bHLH79; r = 0.706–0.868), while negatively correlated with MYB (VIT_04s0008g03720, Target of Myb protein 1; VIT_10s0116g00500, ETC1; VIT_14s0006g01620, MYBC2; VIT_14s0066g01090, MYB24; r =  − 0.844—0.709) and bHLH (VIT_00s0274g00070, bHLH112; VIT_03s0038g04760, bHLH63; VIT_05s0029g00390, bHLH77; VIT_05s0077g00750, bHLH147; VIT_08s0040g01240, bHLH106; VIT_11s0016g03560, bHLH68; VIT_19s0014g04670, bHLH123; r =  − 0.962—0.702) (Data S3). Light-independent anthocyanin synthesis was positively correlated with MYB (VIT_16s0050g02530, TRY; r = 0.882) and bHLH (VIT_05s0029g00390, bHLH77; VIT_09s0002g04120, bHLH68; r = 0.783–0.873), but it negatively correlated with other MYB members (VIT_06s0004g00570, MYB5B; VIT_07s0005g01950, MYB108-like protein 2; VIT_09s0054g01620, PHL6; XLOC_013017, MYB-like protein H isoform X1; r =  − 0.939—0.700).

To evaluate the validity of deep-sequencing data, 4 genes were selected for examination by qRT-PCR, of which only PAL (VIT_08s0040g01710) in A1 was not consistent with that of deep-sequencing, indicating the reliability of high-throughput data (Fig. 6).

Figure 6 qRT-PCR validation of four candidate genes related to anthocyanins synthesis from the different bagging treatments.

qRT-PCR validation of four candidate genes related to anthocyanins synthesis from the different bagging treatments (A, C, E and G) indicate relative expression of PAL(VIT_08s0040g01710), F3H(VIT_18s0001g14310), CHS(VIT_05s0136g00260) and F3′5′H (VIT_06s0009g02970) at S2, respectively. (B, D, F and H) indicate relative expression of PAL(VIT_08s0040g01710), F3H(VIT_18s0001g14310), CHS(VIT_05s0136g00260) and F3′5′H (VIT_06s0009g02970) at S3, respectively. The left y-axis indicates relative gene expression levels were determined by qRT-PCR and analyzed using 2−ΔΔCT Method. The x-axis indicates different treatments. All qRT-PCR for each gene used three biological replicates, with three technical replicates per experiments. Error bars indicate ±SE, and the asterisks represent significant differences from the control, with P < 0.05(∗) or P < 0.01(∗∗).

Discussion

Complete darkness decreased the concentration and composition of total anthocyanins, delayed the veraison, but did not affect the coloration of grape berry skin during harvest

Light is an important factor which affects fruit development and it also participates in a series of processes such as fruit morphological development, quality formation and changes in molecular structure (Feng et al., 2013; Takos et al., 2006a; Takos et al., 2006b). Fruit color is an important parameter for ripening and quality evaluation of fresh and processed fruit (De et al., 2017; Henwood et al., 2018). In the present research, the veraison of B was delayed for 3 days compared with CK whiles the veraison of C and D was delayed for 10 days. The composition of anthocyanins in berries affects the color tone and color stability of resulting wines (Rustioni et al., 2013). Also, the results of this study suggested that the components of anthocyanins decreased gradually with decreases in the light transmittance. All the 24 anthocyanin components detected were present in both CK and A, while Cyanidin-3-O-coumaroylglucoside (cis) was not detected in C, Cyanidin-3-O-coumaroylglucoside (cis), Peonidin-3-O-coumaroylglucoside (trans) and Malvidin-3-O-coumaroylglucoside (trans) were not detected in D from the three development stages. The effect of fruit exposure to light has been investigated in previous studies (Jutamanee & Onnom, 2016; Llorente et al., 2016). As observed in apple (Honda et al., 2017; Honda & Moriya, 2018) and pear (Feng et al., 2010), the bagged fruits displayed no color, but the pericarp turned red when exposed to sunlight. In the absence of light, the anthocyanin synthesis from berry skin of grape cv. ‘Jingxiu’ was inhibited, while the grape cv. ‘Jingyan’ remained unchanged (Zheng et al., 2009). Similarly, although different fruit bags significantly reduced the concentrations of total anthocyanins in our research, the appearance of the berry skin in bagged berries was not different from that of the CK at harvest stage, suggesting that grape varieties respond differently to light (Cortell & Kennedy, 2006). In summary, complete darkness decreased the concentration and composition of total anthocyanins, delayed the veraison, but did not affect the coloration of the berry skin during harvest in grape cv. ‘Marselan’.

Light-independent anthocyanin is dominant for berry skin coloration in grape cv. ‘Marselan’

Fruit-zone shading reduced total anthocyanin accumulation and decreased the 3′-hydroxylated anthocyanin concentration but increased the 3′,5′-hydroxylated anthocyanin concentration in ‘Nebbiolo’ grapes (Price et al., 1995). Similarly, the accumulation of 3′,4′,5′-hydroxylated anthocyanin in grape cv. ‘Yan-73’ decreased under dark conditions, while the concentration of 3′,4′-hydroxylated anthocyanin increased (Guan et al., in press). Studies have shown that trans-structured substances are more stable than cis-structured substances (Schieber & Carle, 2005), and changes in the external environment usually alter the concentration of these substances (Spanos & Wrolstad, 1990; Versari et al., 2001). In grape cv. ‘Redglobe’, both UV-C and refrigeration treatment increased the concentration of cis- and trans-piceid (Crupi et al., 2013). The concentration of trans-resveratrol increased when the temperature increased, while the concentration of cis-resveratrol decreased (Kolouchová-Hanzlı Ková et al., 2004). In our work, Malvidin-3-O-coumaroylglucoside (trans), which showed a significant positive correlation with the total concentration of anthocyanins at the harvest stage and were not detected in D, were considered to be light-induced anthocyanin. These was trans-structures with good stability. Other anthocyanins which were both synthesized in CK and D were considered to be light-independent anthocyanins. The cis-structures of light-independent anthocyanins, Peonidin-3-O-coumaroylglucoside (cis) and Malvidin-3-O-coumaroylglucoside (cis), were involved in light-independent anthocyanins. Therefore, these results indicated that the intensity of light could change the isomerization level of anthocyanins in grapes. In addition to the light-inducible anthocyanins which accounted for 13% in CK, the proportion of light-independent anthocyanins accounted for 87% (Fig. 3A). Therefore, light-independent anthocyanins are the dominant for berry skin coloration in grape cv. ‘Marselan’.

The synthesis of light-inducible anthocyanins and light-independent anthocyanins may be regulated by specific genes

Anthocyanin synthesis belongs to the flavonoid metabolic pathway, which involves a series of metabolic reactions and intermediates synthesis, resulting in different components of anthocyanins (Matsui et al., 2016). Shading and bagging experiments showed that light could stimulate the up-regulation of anthocyanin synthesis related genes, thereby increasing anthocyanin accumulation in fruits (Guan et al., 2016; Zhang et al., 2016). Studies have shown that PAL gene expression is responsive to light (Liang et al., 1989; Huang et al., 2010; Zhu et al., 2018). Similarly, the results of the present study revealed that the expression levels of PALs (VIT_16s0039g01360, VIT_00s2849g00010, VIT_16s0039g01240, VIT_08s0040g01710, VIT_16s0039g01300) and F3H (VIT_18s0001g14310) which positively correlated with the concentration of Malvidin-3-O-coumaroylglucoside (trans), were considered to positively regulate the synthesis of light-inducible anthocyanins. Zhang and others also found that over-expression of the CHS gene enhanced high light resistance by synthesizing more anthocyanins in Arabidopsis leaves (Zhang et al., 2017). The expression of CHS (VIT_05s0136g00260) which was negatively correlated with the concentration of Malvidin-3-O-coumaroylglucoside (cis) in the present study was considered to be related to the synthesis of light-independent anthocyanins. The accumulation of 3′,4′,5′-hydroxylated and 3′,4′-hydroxylated anthocyanins respond differently to dark conditions and was related to the expression levels of VvF3′H and VvF3′5′H (Guan et al., in press). Similarly, the expression levels of CHS (VIT_14s0068g00920), F3H (VIT_04s0023g03370) and F3′5′H (VIT_06s0009g02970) which negatively correlated with the concentration of Malvidin-3-O-coumaroylglucoside (trans) in our work, were considered to negatively regulate the synthesis of light-inducible anthocyanins. In addition, the expression of F3′5′H (VIT_06s0009g02840) which positively correlated with the concentration of Malvidin-3-O-coumaroylglucoside (cis), was considered to positively regulate the synthesis of light-independent anthocyanins.

Members of the MYB transcription factor families, particularly, MYBA1, MYBA2, MYB5A, MYB5B, MYBPA1 and MYBPA2, were involved in the regulation of structural genes in the flavonoid pathway (Ali, 2011; Cavallini et al., 2014). The expression of R2R3-MYB transcription factor MYBF1 was light-inducible, implicating MYBF1 in the transcripts regulation of FLS in grape (Czemmel et al., 2009). MYB10 was most responsive to light while the transcripts declined to undetectable levels in the fruit preserved in the dark (Daniela et al., 2013). In this study, light-inducible anthocyanin synthesis positively correlated with MYBC2-L1, MYBA2, MYBA1, MYBA3, MYB30, bHLH121, bHLH30 and bHLH79, but it negatively correlated with target of MYB protein 10, ETC1, MYBC2, MYB24, bHLH112, bHLH63, bHLH77, bHLH147, bHLH106, bHLH68 and bHLH123. Light-independent anthocyanin synthesis was positively correlated with TRY, bHLH77 and bHLH68, but it negatively correlated with MYB5B, MYB108-like protein 2, PHL6 and MYB-like protein H isoform X1.

Conclusion

Light could affect the accumulation of light-inducible anthocyanins by positively regulating the expression levels of PALs, F3H (VIT_18s0001g14310), MYBC2-L1, MYBA2, MYBA1, MYBA3, MYB30, bHLH121, bHLH30 and bHLH79, while negatively regulating the expression of CHS (VIT_14s0068g00920), F3H (VIT_04s0023g03370), F3′5′H (VIT_06s0009g02970), target of Myb protein 10, ETC1, MYBC2, MYB24, bHLH112, bHLH63, bHLH77, bHLH147, bHLH106, bHLH68 and bHLH123. On the contrary, darkness may promote the expression of F3′5′H (VIT_06s0009g02840), TRY, bHLH77 and bHLH68 but inhibit the expression of CHS (VIT_05s0136g00260), MYB5B, MYB108-like protein 2, PHL6 and MYB-like protein H isoform X1, and therefore promote the synthesis of light-independent anthocyanins.

Supplemental Information

Supplemental Information 1 Supplemental Material dataset and tables

Correlation analysis

Click here for additional data file.

Supplemental Information 2 Supplementary tables

Primer pairs used for qRT-PCR and RNA-Seq data

Click here for additional data file.

Supplemental Information 3 Venn diagram of numbers of DEGs in different bagging treatments: CK1 versus A1, CK1 versus B1, CK1 versus C1 and CK1 versus D1 (S2),CK2 versus A2, CK2 versus B2, CK2 versus C2 and CK2 versus D2 (S3)

Click here for additional data file.

Additional Information and Declarations

Competing Interests

Author Contributions

Data Availability

The authors declare there are no competing interests.

Zong-Huan Ma and Wen-Fang Li performed the experiments, analyzed the data, contributed reagents/materials/analysis tools, prepared figures and/or tables, authored or reviewed drafts of the paper, approved the final draft.

Juan Mao performed the experiments, approved the final draft.

Wei Li contributed reagents/materials/analysis tools, prepared figures and/or tables, approved the final draft.

Cun-Wu Zuo analyzed the data, prepared figures and/or tables, authored or reviewed drafts of the paper, approved the final draft.

Xin Zhao analyzed the data, contributed reagents/materials/analysis tools, approved the final draft.

Mohammed Mujitaba Dawuda and Xing-Yun Shi approved the final draft.

Bai-Hong Chen conceived and designed the experiments, approved the final draft.

The following information was supplied regarding data availability:

The raw data are available in the Supplemental Files.

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
