# Peer review of "Synthesis of light-inducible and light-independent anthocyanins regulated by specific genes in grape ‘Marselan’ (V. vinifera L.)"

_PeerJ, doi:10.7717/peerj.6521_

## Round 0.1 · original submission · Minor Revisions

Please revise also the statistics. Consider elaborating heatmap figures with R software. Provide a detailed response to each reviewer.

Reviewer 1 ·

Basic reporting

.

Experimental design

.

Validity of the findings

.

Additional comments

The subject of the paper is very interesting. The article is generally well written and will serve as a valuable resource for our knowledge. I suggest going ahead in a future in such a research line focusing on other bioactive ingredients in plant foods.

Anyway, I feel the authors should comment on some of the related papers on phenolic bioactive compounds in plant foods, such as:
-. “Pattern recognition of three Vitis vinifera L. red grapes varieties based on anthocyanin and flavonol fingerprints, with correlations between their biosynthesis pathways. Food Chemistry, 2012, 130, 9-19”.
-. “Anthocyanins and flavonols berries from Vitis vinifera L. cv. Brancellao separately collected from two different positions within the cluster. Food Chemistry, 2012, 135(1), 47-56”.
-. “Flavonoids in Gran Negro berries collected from shoulders and tips within the cluster, and comparison with Brancellao and Mouratón varieties. Food Chemistry, 2012, 133(3), 806-815”.
-. “Evolution of flavonoids in Mouratón berries taken from both bunch halves. Food Chemistry, 2013, 138(2), 1868-1877”.
-. Relationship between the sensory-determined astringency and the flavanolic composition of red wines. Journal of Agricultural and Food Chemistry, 2012, 60(50), 12355-12361”.
-. “Increasing the added-value of onions as a source of antioxidant flavonoids: a critical review. Critical Reviews in Food Science and Nutrition, 2014, 54(8), 1050-1062”.
-. “Comprehensive identification of walnut polyphenols by liquid chromatography coupled to linear ion trap-Orbitrap mass spectrometry. Food Chemistry, 2014, 152, pp. 340-348”.
-. “A critical review of bioactive food components, and of their functional mechanisms, biological effects and health outcomes. Current Pharmaceutical Design, 2017, 23(19), 2731-2741 (doi: 10.2174/1381612823666170317122913).”
-. “A critical review of the characterization of polyphenol-protein interactions and of their potential use for improving food quality. Current Pharmaceutical Design, 2017, 23(19), 2742-2753 (doi: 10.2174/1381612823666170202112530).”

The science presented is of a high quality with plenty of detail. This is a substantial manuscript that occupies the bounds between food chemistry and analytical chemistry. I do express my positive opinion on the acceptance of the article to be published after minor revision.

Reviewer 2 ·

Basic reporting

1) The English writing should be thoroughly improved. Some of the sentences were so difficult to understand that I could not even guess what the author means. In particular, from library construction to quantitative real time PCR (lines 168 to 230). Some of the terms are not standard, like 'cDNA database' instead of 'cDNA libraries' etc. Most of error prone ones were marked in yellow in details, some were with comments.
2) Carefully check the spellings, like cultivar name 'Marselan' in title page or 'Marselan' in the title, but 'Matheran' in main body context, even in the abstract.
3) Figures were well described and labeled appropriately in the manuscript. All data were appropriate presented too.

Experimental design

Authors investigated the anthocyanin component individuals in berry skins with various light intensity by bagging experiment. The results were interesting and can bring forward our understanding about the relationship between light dependent and light independent synthesized pigments. Coupling with high throughput sequencing, it is expecting to get some useful knowledge in the area.
However, due to bad english writing, I cannot fully understand the details of sample collection of the sequencing samples.

Validity of the findings

Some conclusions should be carefully drawn and it is too speculative in the current version. Examples like:
In the 'Effects of light intensity on anthocyanin composition in grape berry skin' section (line 263 to 265), Pn-3-o-cou and Mv-3-o-cu may be induced by light intensity. This conclusion was not fully supported by the data in suppl. dataset s1. Mv-3-o was not detected in D of S3 stage. But when compare the quantity from CK, A, B, C, the light intensity decreasing, the content of this kind of pigment did not show constant pattern of been light inducible. Again, CHS was considered negatively regulate the synthesis of non-light-inducible anthocyanins (line 452). This conclusion could not be accepted, especially authors only investigate the expression of CHS (a family sometimes with several members, which was not specify or deeply investigated here) with only Mv-3-O-cou. Authors should draw more stringent conclusions.

Additional comments

Detailed concerns
Major points, 1) most of the conclusions were based on the results from table s1, so I suggest you put the table in the main text instead of in supplementary data. In considering that it is a large table, other presentation forms like heatmaps can be applied. 2) Details about the samples collection, replications, corresponding to the library construction should be clear. 3) I do not know how did the authors picked out MYB and bHLH TFs and why they chose the ones listed in the text. If all TF members were included, it is obviously not suitable; if particular ones were selected, put some background information in the introduction section. 4) conclusions should be carefully drawn, for example, Mv-3-O-coumaroylglu (trans) may be induced by light intensity (line 264), why its content decrease when light intensity increase (from table S1)?

line 51, there are more suitable references than that of tomato SB. It is an anthocyanin mutant, not a typical one.
The introduction section from line 48 to 71 and from line 72 to 88 seems to lack logic behind. Recommendations are light intensity and light quality affections, structural genes or regulatory components in either section.
line 130, please add information about why you choose S2 and S3.
line 136, clarify why not choose the collected skins, but from treated plants and explain how you get the fine powder.
line 191, if indexs were used, please indicate the sequence.
line 255, I strongly suggest that Supplementary dataset S1 was used as main results table or as heatmap used in the main text.
line 277, since Pn-3-O-coumaroylglu (trans) only draw from 1% to 0, it is not convincing to get the conclusion
line 290, if your libraries were sequenced 100 bp in length (line 196), how do you get 150 bp reads?
line 295, please provide more information about how do you conducted pairwise comparison.
line 300, we know that Go terms can category genes into three classes, and this was not what we want to know about the data. You should provide detailed information in the three classes, for example, which special go terms in BP or MF or CC category were enriched. That's what authors expected.
line 450 to 452, CHS was negatively regulate the synthesis of non-lint-inducible anthocyanins, it is too speculative to get this conclusion.

Annotated reviews are not available for download in order to protect the identity of reviewers who chose to remain anonymous.

---

## Round 0.2 · accepted · Accept

Dear author

I can read that you have addressed all the reviewers concerns and that you have added the heatmap figure. The reviewers comments have been responded adequately and your manuscript has improved considerably. I congratulate you for the nice piece of work, which will add value to PeerJ.